# Controlled noninvasive modulation of deep brain regions in humans

Thomas Riis [1✉], Daniel Feldman [2], Brian Mickey [1,2] & Jan Kubanek [1✉]

Transcranial focused ultrasound provides noninvasive and reversible approaches for precise and personalized manipulations of brain circuits, with the potential to transform our understanding of brain function and treatments of brain dysfunction. However, effective applications in humans have been limited by the human head, which attenuates and distorts ultrasound severely and unpredictably. This has led to uncertain ultrasound intensities delivered into the brain. Here, we address this lingering barrier using a direct measurement approach that can be repeatedly applied to the human brain. The approach uses an ultrasonic scan of the head to measure and compensate for the attenuation of the ultrasound by all obstacles within the ultrasound path. No other imaging modality is required and the method is parameter-free and personalized to each subject. The approach accurately restores operators' intended intensities inside ex-vivo human skulls. Moreover, the approach is critical for effective modulation of deep brain regions in humans. When applied, the approach modulates fMRI Blood Oxygen Level Dependent (BOLD) activity in disease-relevant deep brain regions. This tool unlocks the potential of emerging approaches based on low-intensity ultrasound for controlled manipulations of neural circuits in humans.

---

[1] Department of Biomedical Engineering, University of Utah, Salt Lake City, UT 84102, USA. [2] Department of Psychiatry, University of Utah, Salt Lake City, UT 84102, USA. ✉email: tom.riis@utah.edu; jan.kubanek@utah.edu

Transcranial low-intensity ultrasound provides a new set of methods to noninvasively and reversibly manipulate neural circuits[1,2]. The approaches have included transient[3–5] and durable[6–11] modulation of neural circuits, and the delivery of specific drugs across the intact[12–15] and transiently opened[16,17] blood-brain barrier. Unlike other noninvasive approaches, ultrasound-based methods reach millimeter-level precision deep in the brain[18]. Since these approaches are noninvasive and reversible, they provide flexible, systematic tools for causal mapping of brain function and personalized diagnostic and therapeutic protocols.

However, the effectiveness and safety of these emerging approaches have been hampered by a formidable barrier: the acoustically complex human head. The human skull alone attenuates the ultrasound by a factor of 4.5–64 depending on individual and skull segment[19–21]. Hair[22,23], acoustic coupling to the head[24,25], and entrapped bubbles or air pockets between the transducer and the subject's head[26] present additional significant barriers. The joint outcome of these barriers is a severe (Fig. 1a) and highly variable (Fig. 1b) attenuation[27,28], which has precluded the delivery of deterministic ultrasound intensity. The uncertainties about the intensities delivered into the brain have severely limited emerging reversible therapeutic applications. This is because these approaches—including neuromodulation and drug delivery—are sensitive to the ultrasound intensity and operate within a narrow window of effectiveness and safety[13,29–31].

Current methods to address the ultrasound attenuation by the head are either invasive or not safe for routine applications in humans. For instance, the ultrasound intensity delivered into the brain could be measured using receivers implanted in the brain[32,33] or using microbubbles injected into a person's blood stream[34–36]. The invasiveness of these methods has limited their deployment. Noninvasive imaging approaches based on MRI, including thermometry and acoustic radiation force imaging[18,37,38], require high ultrasound intensities to heat up or mechanically push on a target in the brain. This has limited applications to ablative brain treatments[18]. Computed Tomography (CT) scans of the head have thus far also been used predominantly for ablative brain treatments[18]. CT scans can be used to estimate the ultrasound dephasing by the skull[18,39,40]—which is important for maximizing the delivered intensity for an effective ablation, but this ionizing form of energy has been less successful in measuring the attenuation of ultrasound by the skull[41,42] and is incapable of accounting for the attenuation by scalp and acoustic coupling. These are crucial limitations for the delivery of controlled intensity for reversible therapies in which the ultrasound attenuation constitutes the key factor (Fig. 1).

To address this lingering barrier, we have developed an approach that directly measures and compensates for all obstacles in the ultrasound path and can be routinely applied to the human brain. The approach, Relative Through-Transmit (RTT), is based on ultrasound—the same form of energy and frequency as that used for the ensuing interventions. RTT applies a low-intensity ultrasound pulse through each segment of the head to directly measure its acoustic attenuation, and compensates for these measured values prior to performing an intervention. Because RTT measures the attenuation directly, it does not require other scans of the head or free parameters. We found that RTT accurately restores transcranial intensities that operators intend to deliver into the brain. Furthermore, RTT enabled effective and safe modulation of deep brain circuits in humans.

## Results

### The attenuation barrier for controlled transcranial ultrasound.

Figure 1 a demonstrates the severity of the attenuation of ultrasound by the skull. Across 8 ex-vivo skulls, we found that the ultrasound intensity delivered into a deep brain location (see Methods) is attenuated by a factor of $11.4 \pm 6.8$ (mean ± s.d.), which replicates previous findings both in terms of the magnitude and the variability of the attenuation[19]. In principle, the attenuation

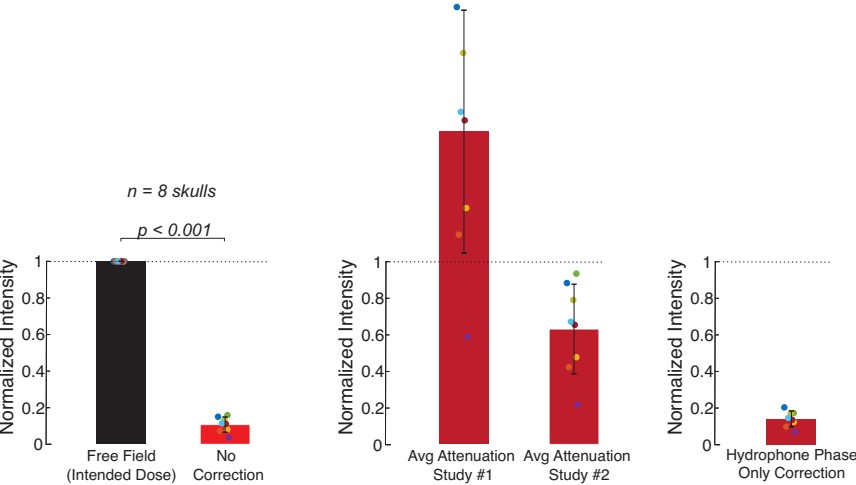

**a** Ultrasound attenuation due to skull **b** Correction for attenuation using average tabulated values **c** Correction for skull using perfect phase correction

**Fig. 1 The ultrasound skull attenuation problem. a** Spatial peak intensity (mean ± s.d.) of the measured field delivered through 8 ex-vivo human skulls into a central target, separately for the intended intensity (black bar) and the intensity following the propagation of ultrasound through the skull (red bar). The intensity field was measured using a calibrated hydrophone (see Methods). **b** The acoustic attenuation by the skull could be estimated using tabulated values (e.g., reference #1[20] or #2[21]), but the high variability of the attenuation across individuals makes such estimates inaccurate and uncertain. Same ex-vivo skulls as in **a**. **c** The problem cannot be addressed using the many existing methods that correct for the ultrasound dephasing. Same skulls as in panels a and b, following the hypothetical ideal correction for the ultrasound dephasing using ground-truth hydrophone phase measurements obtained at the target.

**Fig. 2 Approach for controlled delivery of ultrasound into the brain.** Problem (red): The human skull precludes the delivery of controlled ultrasound dose into the brain. The human skull attenuates ultrasound strongly and unpredictably, leading to low and variable intensity delivered into a brain target. This is illustrated by the red target inside an MRI scan of a human subject and the red bar representing the actual intensity electronically focused into a central target through an ex-vivo skull. Solution (green). We developed a method, relative through-transmit (RTT), which uses brief through-transmit pulses of low-intensity ultrasound to measure the ultrasound aberrations by the skull. The top waveform is an example of a typical through-transmit (Tx) signal recorded (Rx) through the head of a participant with unshaved hair. The scan is performed with a subject's head in the ultrasound path, and the obtained signals are compared with reference signals that had been obtained in water (bottom signal). From the relative differences in the magnitudes and times of flight of the received signals, RTT computes the attenuation and phase shift of each skull segment within each ultrasonic beam. These values are then used to scale up and delay the emission of ultrasound from the individual elements and thus compensate for the skull (Suppl. Fig. 2), restoring the intended intensity at the target (right bars; green). No CT or MRI images of the head are required. The green bar shows the mean ± s.e.m. corrected intensity across 8 ex-vivo skulls and 3 targets evaluated in detail below (center, 10 mm axial, 10 mm lateral).

could be estimated using tabulated values (e.g., ref. [20,21]), but the high variability of the attenuation across individuals makes such estimates inaccurate and uncertain (Fig. 1b). For instance, using the values of those two studies would over- and under-estimate the average value by a factor of $1.7 \pm 0.67$ and $0.63 \pm 0.25$, respectively, and lead to high variability (pooled standard deviation equal to 0.47 with respect to normalized intensity of 1.0). A compensation for the dephasing of the ultrasound, which can be obtained using many existing methods[18,32–36,39,40,43–45], is useful for maximizing the delivered intensity and thus for ultrasound-based surgeries[18]. Nonetheless, even the hypothetically ideal correction for the phase (Fig. 1c) leaves a discrepancy of 85% between the intended and actual intensities delivered into a brain target. Therefore, existing methods aimed at correcting for the dephasing of the ultrasound are insufficient to account for the delivered intensity; a compensation for the attenuation is required (Suppl. Fig. 1).

**Noninvasive approach to effectively account for the attenuation**. To address this critical issue, we have developed an ultrasound-based approach, RTT, which uses the same energy and the same frequency as those for the ensuing therapies. This concept enables direct measurements of the attenuation (and dephasing) of all obstacles within the ultrasound path, including the skull, the hair, and the coupling between the scalp and the transducers. RTT subsequently compensates for the measured aberrations and thus delivers into a transcranial target a defined level of ultrasound intensity. Specifically, RTT rests on two sets of transducer-phased arrays positioned at opposite sides of the head (Fig. 2). Each element of the array operates in both transmit and receive modes. This complete through-transmit system enables ultrasound-based compensation for the attenuation (and dephasing) of each ultrasonic beam (Fig. 2).

**Transcranial delivery of controlled intensity**. We implemented RTT using two sets of high-element-count arrays, and connected the arrays to a driving system that independently transmits

signals from and listens to each element (Methods). We then used the hardware to perform RTT scans (Methods) through human ex-vivo skulls. Both RTT and the ensuing ultrasound delivery harness the same system; no additional hardware or imaging modality is necessary. The ex-vivo skulls enabled us to measure the transcranial fields using a calibrated hydrophone (see Methods) and thus validate the accuracy of the RTT compensation. We evaluated the measured intensities in four conditions. First, we measured the intensities in the free field, which presents no obstacles for ultrasound. This intensity corresponds to the intensity intended to be delivered into the target by the operator. Second, we introduced skulls between the hardware and the hydrophone, and measured the resulting intensities. This case presents the worst-case, yet commonly performed, scenario of no correction for the skull. Third, we evaluated the best possible scenario, the hypothetical ideal correction for the skull. To do that, we used the hydrophone to measure the attenuation and dephasing for each element of the device, thus obtaining ground-truth values. And fourth, we applied the RTT correction.

We performed these measurement inside 8 water-immersed, degassed human ex-vivo skulls. Figure 3 shows the spatial peak intensity and the associated field for a target positioned at the center of the two transducers. The figure reinforces the notion that human skulls severely dampen the intensity delivered into the brain (red). Compared with the free-field values, the ultrasound intensity through the skulls was attenuated by a factor of $11.4 \pm 6.8$ (mean ± s.d.), degrading it to $10.7 \pm 4.2\%$ of the intended intensity. The difference between the free-field and through-skull values was significant ($t_7 = 60.7$, $p = 8.6 \times 10^{-11}$, paired two-tailed t-test).

We then applied RTT using the phased arrays. Figure 3 shows that RTT effectively restores the intended target intensity (green). The RTT-compensated intensity constituted $98.8 \pm 17.8\%$ (mean ± s.d.) of the intended values in free-field, and there was no significant difference between the mean of two conditions ($t_7 = 0.18$, $p = 0.86$). The average value was also not significantly different from the hypothetical, best-possible correction based on

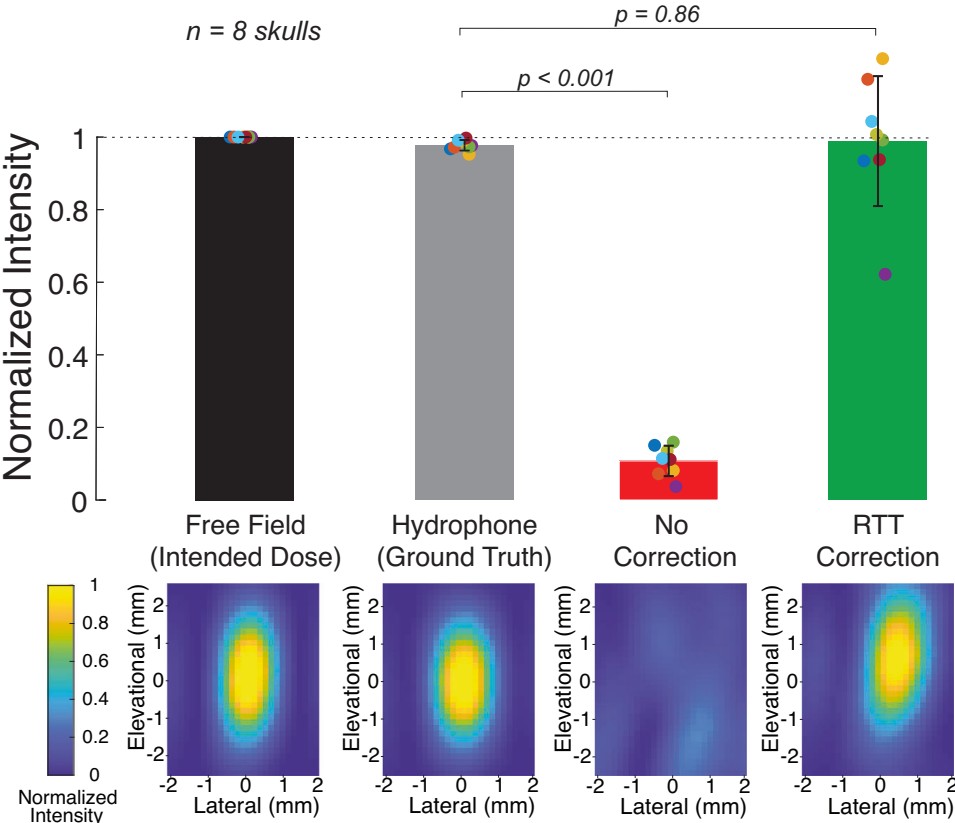

**Fig. 3 Relative Through-Transmit (RTT) compensates for the skull and restores the intended intensity at target.** Ultrasound fields obtained at a central target inside ex-vivo human skulls ($n = 8$), separately for the hypothetical ideal correction (gray), no correction (red), and RTT (green). The bars show the spatial peak intensities (mean ± s.d.) of the field for each case. The 2D images corresponding to each bar show an example spatial distribution of the ultrasonic fields at the central target. Data for off-center targets are provided in Suppl. Fig. 6.

the hydrophone ground-truth measurements inside the skull (black bar; $t_7 = 0.17$, $p = 0.86$, paired two-tailed t-test). One skull (purple datapoint) attenuated the ultrasound severely (a factor of 26.9 attenuation). This was likely be due to visually present outgrowths possibly related to hyperostosis, as assessed by a neurosurgeon. For this skull, the RTT correction was less accurate, attaining 62.0% of the intended intensity.

**Robustness of the approach.** We tested the robustness of RTT with respect to specific hardware. In particular, we additionally implemented RTT on arrays that had the same number of elements but much larger aperture (Suppl. Fig. 3). For this configuration, skulls ($n = 4$ specimens) degraded the intensity at the geometric center to $6.3 \pm 1.7\%$ of the intended, free-field value, in line with Fig. 1. The RTT compensation recovered the intensity at the target to $104 \pm 18.1\%$ of the intended value. Following the compensation, there was no significant difference between the intended and mean RTT-recovered intensities ($t_3 = 0.47$, $p = 0.67$, paired two-tailed t-test). Therefore, RTT is robust with respect to particular hardware implementation.

We further tested the robustness of RTT with respect to brain target location. To do so, we used the phased arrays to refocus the ultrasound into targets within the steering range of the ultrasonic arrays: targets 10 mm axial, 20 mm axial, 10 mm lateral, 20 mm lateral, and 15 mm elevational with respect to the central target (Suppl. Fig. 6). RTT correction brought the delivered intensity to $96.3 \pm 21.4\%$, $94.8 \pm 23.2\%$, $92.8 \pm 16.4\%$, $62.5 \pm 15.7\%$, and $71.6 \pm 18.03\%$ of the intended value in each target respectively. There was no statistical difference between the mean of intended peak intensities and the RTT-compensated peak intensities at the

central target ($t_7 = 0.18$, $p = 0.86$, paired two-tailed t-test), 10 mm axial ($t_7 = 0.48$, $p = 0.64$), 10 mm lateral ($t_7 = 0.42$, $p = 0.55$), and 20 mm axial ($t_7 = 1.23$, $p = 0.26$). There was a significant difference in the average delivered intensity at the target 20 mm lateral ($t_7 = 6.748$, $p = 0.0002$) and 15 mm elevational ($t_7 = 4.5$, $p = 0.003$).

**Correction for attenuation and dephasing.** We validated the notion of the relative contribution of the two key components of the ultrasound aberration by the skull—the attenuation and the dephasing. Suppl. Fig. 1 shows the spatial peak intensity following the engagement of each correction type in isolation as well as their joint application. At the central target, phase-only correction resulted in an average intensity of $13.8 \pm 4.3\%$ (mean ± s.d.) for the ideal hydrophone correction (gray) and $11.9 \pm 4.9\%$ for RTT (green). The no correction value (red) was $10.7 \pm 4.2\%$ of the free field intensity. The amplitude-only RTT correction brought the peak spatial intensity to $93.8 \pm 28.9$. The inclusion of the correction for the phase is not necessary but additionally beneficial in that the resulting joint correction (Suppl. Fig. 1c) brings the average delivered intensity to $98.8 \pm 17.8\%$.

**Delivery through the human head.** We next assessed whether RTT could be applied to the human head, which presents additional key barriers for transcranial ultrasound, including the scalp, hair, and acoustic coupling (Suppl. Fig. 4). Under approved IRB protocols, we applied RTT through the head of 6 human subjects with hair. Suppl. Fig. 5 (blue) shows the average through-transmit attenuation through both sides of the head, separately for each subject. The figure demonstrates that RTT offers

through-transmit measurement capacity comparable to the previously evaluated ex-vivo skulls (gray). Specifically, the receiving elements on the opposite side of the head (Fig. 2) recorded an average of $7.2 \pm 4.4\%$ (mean ± SD, $n = 6$ subjects) of the signal amplitude when RTT was applied through the human skull, and $12.6 \pm 8.2\%$ for the characterized ex-vivo human skulls (mean ± SD, $n = 8$ skulls). The additional factor of 1.7 attenuation is expected because the application of ultrasound through the human head incurs additional attenuation by hair, scalp, coupling, bubbles or air pockets in between, as well as tissues inside the skull. RTT measures and compensates for the impact of all these obstacles.

RTT was designed to be safe. The RTT scan consists of brief (<100 µs) low-intensity (average peak pressure of 80 kPa in free field; Suppl. Fig. 10) pings of ultrasound. A full RTT scan takes less than one second to complete. Subjects ($n = 6$) did not feel any discomfort during the procedure. No subject reported side effects at 1-week follow-up. Thus, RTT can be safely applied to the head of humans and directly measures the attenuation by all obstacles within the ultrasound path.

**Application to neuromodulation in humans.** Neuromodulation with transcranial focused ultrasound has been effective in rodents but robust and reproducible effects in humans have remained elusive[30,46,47]. The human head and acoustic coupling to unshaved hair have been the key barriers. We therefore hypothesized that RTT's ability to compensate for these obstacles could elevate the robustness of ultrasonic neuromodulation in humans. To test this hypothesis, we preregistered a clinical study (NCT05301036) and obtained an IRB approval for applying RTT and ultrasonic stimulation in patients with major depression. We specifically targeted a deep brain region, the subgenual cingulate cortex (SGC), which has been hypothesized to be overactive in depression[48]. The hardware and RTT were administered to the patients in the same way as to the healthy subjects, i.e., without anesthesia or hair shaving (Methods).

RTT detected a substantial and subject-specific attenuation of ultrasound by the head (Fig. 4a). In subject A and B, the procedure compensated for the attenuation by scaling the ultrasound amplitude for the individual elements by a factor of $4.15 \pm 1.08$ and $5.33 \pm 1.10$, respectively. Following this compensation, we delivered ultrasound into the SGC. The targeting was performed under MRI guidance (Methods). The ultrasound was delivered into the target for 10 minutes, interleaving 1 minute of ON stimulation periods with 1 minute OFF periods. The waveform constituted brief pulses (30 ms) delivered every 4 s with 1 MPa peak pressure at target. This way, the time-averaged intensity of 360 mW cm$^{-2}$ safely complied with the FDA 510(k) Track 3 level of 720 mW/cm$^{-2}$ [31].

We measured the engagement of the target using functional MRI, contrasting standard BOLD activity during the ON and OFF epochs of the ultrasound (Methods). We observed a robust engagement of the target by the neuromodulatory ultrasound pulses (Fig. 4b). The targeted region showed a significant BOLD modulation by the ultrasound in Subject A (peak level: $p = 7.79 \times 10^{-11}$, $t = 6.64$, $Z_E = 6.40$, cluster-level $p < 0.001$; False Discovery Rate corrected, $k_E = 2130$ voxels) and Subject B (peak level: $p = 0.006$, $t = 3.94$, $Z_E = 3.84$, cluster-level $p < 0.001$; False Discovery Rate corrected, $k_E = 76$ voxels). The locations of significant BOLD modulation were centered at the ultrasound target (SGC, white circles). In Subject B, there was additional modulation anterior to the SGC, which likely represents a functionally connected circuit.

We pulsed the ultrasound to stay within 50% of the FDA 510(k) Track 3 time-average intensity. As a consequence, the

ultrasound was delivered at a low duty (0.8%). Low-duty sonication generally leads to an inhibition of neural structures[49], and this is indeed what we observed in both subjects (Fig. 4c). The inhibitory effect is particularly suitable for treatments of the SGC circuitry, which is generally hyperactive in patients with major depression[48].

We made sure that the deep brain neuromodulation was not due to auditory artifacts that can be associated with ultrasound. This is unlikely as 1) there was no modulation of auditory regions and 2) the modulation was specifically observed in the targeted region. However, we collected data in two important control conditions. First, we evaluated the effects when the RTT correction was not applied. In this case, we observed no significant effect (Suppl. Fig. 11a; cluster-level $p > 0.05$). Therefore, RTT was critical for the neuromodulation. And second, we included a sham condition. The sham stimulus delivered into the brain the same stimulus parameters and energy but was not focused (Methods). The sham stimulation did not elicit significant modulation (cluster-level $p > 0.05$) of the BOLD within the target (Suppl. Fig. 12).

We ensured that the stimulation levels strictly complied with the FDA 510(k) levels[31]. Indeed, no side effects were reported by the subjects following the stimulation (Suppl. Table S1). We also confirmed that the method did not lead to appreciable heating of the skull. Specifically, we performed simulations and measurements of the temperature rise inside 3 ex-vivo skulls (Methods), following the application of RTT. In both the simulation and the measurements, the ultrasonic stimuli matched the 1 MPa pressure amplitude used in the SGC stimulation (Fig. 4). Fig. 5 shows the simulated and measured peak temperature rise due to ultrasonic stimulation as a function of pulse duration. The pulse duration used in the stimulation of the subjects (30 ms; Fig. 4) led to a maximum of 0.047°C temperature increase across all three skulls tested. The simulations and the measurements did not incorporate heat convection by blood vessels, and therefore likely represent upper-bound estimates of the temperature increase.

## Discussion

We developed a noninvasive approach for repeated deployment in humans that effectively compensates for the aberrations of ultrasound by the head and acoustic coupling. The approach enables operators to control the ultrasound intensity delivered into deep brain targets, and thus opens a path to effective and safe applications of emerging reversible ultrasound-based therapies, including neuromodulation and local delivery of drugs. RTT combines noninvasiveness with effective attenuation compensation while being applied safely to the human head and brain.

The human head has been a formidable barrier for current and emerging applications of ultrasound to the brain. This issue has been particularly limiting for reversible approaches based on transcranial ultrasound, which require the delivery of intensity that is both effective and safe. To address this issue, we developed an ultrasound-based method, RTT, that compensates for the strong and variable attenuation of ultrasound by the head (Fig. 3, Suppl. Fig. 1). We implemented RTT in hardware and showed that it faithfully restores the intensities delivered into individual transcranial targets. Moreover, we found that the approach enabled effective and reproducible modulation of a deep brain region in humans (Fig. 4). No safety concerns were reported by 6 human subjects (Fig. 4, Suppl. Fig. 5, Suppl. Table S1). RTT was found to be robust with respect to ultrasonic hardware, target location, and ultrasound intensity levels.

RTT compensates for both the ultrasound attenuation by the skull (critical; Suppl. Fig. 1) and for the ultrasound dephasing (beneficial; Suppl. Fig. 1). Current FDA-approved treatments use

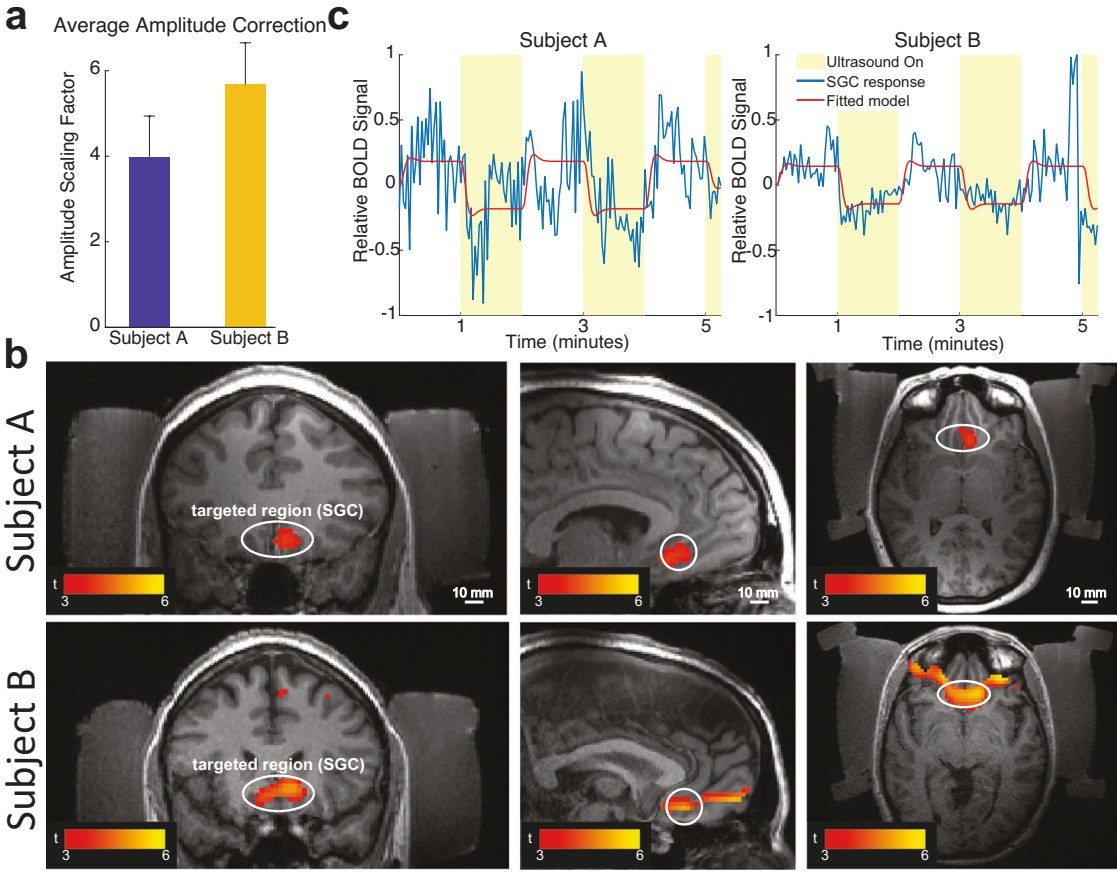

**Fig. 4 Robust engagement of deep brain regions in humans. a** Mean ± s.d. scaling factor across all elements (n=252) that was necessary to compensate for the attenuation of ultrasound by each subject's head and acoustic coupling. There is a notable variability across subjects, elements, and sonication sessions. **b** fMRI Blood Oxygen Level Dependent (BOLD) response to stimulation of the subgenual cingulate (SGC) following RTT correction. The figures show coronal, sagittal, and axial views, respectively. The stimulation and acquisition parameters are provided in Methods. **c** The modulation of the BOLD signal by the ON and OFF ultrasound conditions. The fitted linear model (red) assumes standard hemodynamic response (Methods).

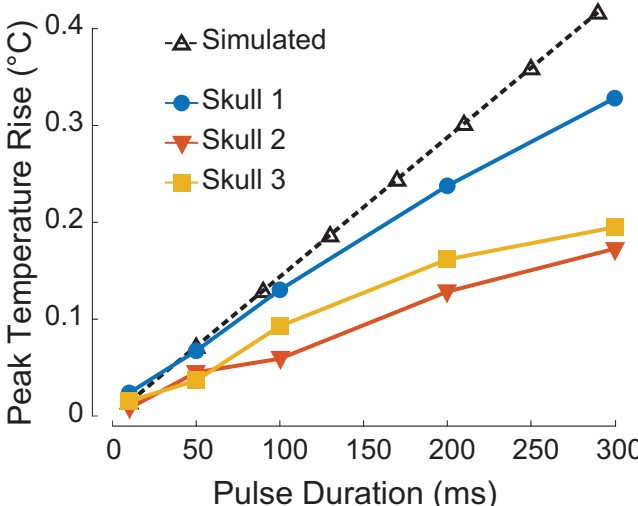

**Fig. 5 Stimulation following Relative Through-Transmit (RTT) is applied through the human skull safely.** Peak skull heating simulated and measured inside three ex-vivo human skulls, as a function of increasing ultrasound pulse durations. The pulses (650 kHz) of each duration were delivered every 4 s for 1 minute at a pressure amplitude of 1.0 MPa at target after RTT correction. The stimulation in the patients (Fig. 4) used 30 ms pulse duration.

high-intensity ultrasound to perform ablative surgeries[48]. For these applications, it is crucial to deliver into the brain target sufficient intensity, which depends on the degree of the ultrasound dephasing by the skull[27,28]. Therefore, the bulk of previous studies regarding correction for skull aberrations of ultrasound have focused on the compensation for the dephasing, and have achieved adequate phase correction[18,32–36,39,40,43–45]. Crucially, however, reversible transcranial applications have a distinct goal —the delivery of deterministic dose that is high enough to ascertain effectiveness, while also low enough to ensure safety. We have found that for this purpose, i.e., the delivery of a predictable dose into a therapeutic target, the correction for the *attenuation* is far more important than the correction for the dephasing (Fig. 1, Suppl. Fig. 1).

RTT is practical in that it takes into consideration all obstacles within the ultrasound path and requires no CT or MRI scans of the head or associated simulations. RTT implements a virtual line path between each transmit-receive pair of elements, and thus measures all forms of attenuation (reflection, absorption, and scattering) of all obstacles along this defined path (Suppl. Fig. 2). Conceptually, RTT could be considered a kind of ultrasound computed tomography[50–54]. Unlike tomography, however, RTT compares signals with the head present to reference signals acquired in water (free field), thus specifically determining the relative attenuation of each ultrasonic beam. The method does not aim to provide absolute attenuation values for each voxel of

the tissue, which would be a much more difficult problem. Critically, in RTT, the acoustic path is measured entirely non-invasively. In comparison, time-reversal methods use implanted receivers[32,33] or injected microbubbles[34–36]. Although these methods can provide accurate corrections, their invasiveness has limited their deployment in humans.

Two previous studies used an ultrasound-based through-transmit correction approach to improve ultrasound imaging through the skull[43,44]. Since the method has been particularly developed for imaging purposes performed with small transducers, this allowed the authors to assume a constant value of phase through the proximal segment of the skull. No such assumption can be made for RTT, which uses large, therapeutic arrays and solves for the distortions in front of each individual element in a system of equations containing all transducer elements.

RTT was applied to the human head safely (Suppl. Fig. 5, Fig. 4, Suppl. Table S1, Suppl. Fig. 13). RTT uses diagnostic-imaging-like, $< 100 \mu s$ pulses that were safely within the FDA 510(k) Track 3 guidelines[31]: $I_{SPTA} = 5.4$ mW/cm$^2$ and $I_{SPPA} = 1.3$ W/cm$^2$. Following the correction, there were no reported side effects in either subject in response to stimulation of the sub-genual cingulate cortex (Suppl. Table S1). The compensation for attenuation could conceivably heat the skull during the ensuing low-intensity application. To address this issue, we calculated and measured the peak skull heating inside ex-vivo human skulls for the respective neuromodulation parameters and found no concerning levels of temperature rise (Fig. 5). This is expected as the overall energy deposited into the skull for low-intensity therapies, after RTT correction, is orders of magnitude lower than the intensities that can produce harmful skull heating[55]. Nonetheless, in every case, including after the application of RTT, it is recommended to pulse the ultrasound such as to follow the $I_{SPTA}$ level of the FDA 510(k) Track 3 guidelines[31]. Furthermore, a previous study[56] performed histology in nonhuman primates and sheep at exposure levels higher than those used in this study. No histological findings were detected[56]. Finally, no tissue damage was detected in T1-weighted MRI images of either subject following the stimulation (Suppl. Fig. 13).

To provide measurements of the skull aberrations that are relevant to the ensuing intervention, both RTT and the intervention should use the same ultrasound frequency. For instance, in the neuromodulatory application evaluated here, both the RTT pings and the neuromodulatory pulses had a 650 kHz carrier frequency.

RTT has certain limitations. First, RTT uses two transducer arrays on opposite sides of the head. This limits applications to either medial-lateral or anterior-posterior directions; correction for a dorsally placed transducer would be difficult to implement. We chose to place transducers parallel over the left and right sides of the skull as there is a through-transmit path with minimal incidence angle to the skull. Second, the method may be limited to frequencies below 2 MHz, which can penetrate both sides of the skull[44]. This is generally not an issue for transcranial therapeutic ultrasound applications, which typically use frequencies below 1 MHz to provide adequate penetration of the skull[18]. Third, the accuracy of RTT was high for electronic treatment envelope of about 20 mm and moderate for broader envelopes. To enable broader treatment envelopes under an accurate correction, the hardware could be physically translated with respect to the head or the method optimized for such purposes. And fourth, the correction for the attenuation was key in delivering a controlled dose within the target; the correction for phase was relatively less effective (Suppl. Fig. 1). In fact, for targets positioned far from the geometric center of the arrays, the phase correction may be detrimental regarding focal volume and target accuracy (Suppl. Fig. 9). Future studies should investigate how the

correction for phase could be applied to off-center regions to improve focal volume and targeting accuracy without the need for repositioning the hardware for each target.

Ultrasonic stimulation can be associated with auditory artifacts[57], which could conceivably influence fMRI BOLD activity. There are four lines of evidence against this possibility in this study. First, there was no activation of the auditory cortex. Second, significant modulation of fMRI BOLD was specifically observed at the ultrasound target. There was no activation of a broad network (e.g., the default mode network). Third, there was no effect during active sham, unfocused stimulation (Suppl. Fig. 12). And fourth, there was no effect during focused stimulation when RTT was not applied (Suppl. Fig. 11).

The outcome of ultrasonic neuromodulation strongly depends on the stimulus intensity. For instance, for a fixed duty cycle value, low intensities generally inhibit, whereas high intensities tend to excite neural circuits[29,30,46,47,49,58]. The controlled intensity provided by RTT therefore opens new possibilities for selective inhibition or excitation of specific brain regions.

Clinical applications will benefit from the method's rapid deployment, which enables operators to check coupling quality before and many times during an ultrasound delivery session, thus accounting for subjects' possible movements and introduction of air gaps or bubbles along the beam path. To maximize the effectiveness and safety of clinical applications, additional work on the compensation could further tighten the confidence on the delivered intensity. This would allow operators to deliver higher intensities without the risk of exceeding safety limits. Clinical applications would also benefit from MRI-based measurements of the in-situ intensities using acoustic radiation force imaging[38].

In summary, we developed and deployed in humans a non-invasive approach that accurately and safely compensates for the severe and unpredictable attenuation of ultrasound by the head. The approach delivers controlled ultrasound intensity through the human skull and enabled targeted modulation of deep brain circuits in humans. This practical correction method is deployed in subjects in real time, does not require CT or MRI scans of the head, and accounts for all obstacles in the ultrasound path, thus circumventing the need for hair shaving. This way, the method is expected to be applicable to a broad spectrum of subjects and patients, enabling precise and personalized diagnoses and treatments of the brain.

## Methods

**Ultrasonic hardware**. RTT was implemented on two hardware platforms. Both systems used two spherical arrays mounted to a rigid plastic frame such that they were positioned opposite to each other and separated by a distance of 180 mm. The array elements of both systems were made of the PMN-PT material, had a surface area of 6 mm x 6 mm, and operated at a fundamental frequency of 650 kHz.

The first system, which was used for the data collection, consisted of two spherically focused arrays (radius of 165 mm; 126 elements; 9 x 14 element grid, inter-element spacing of 0.5 mm). Each array had a height of 55 mm and a width of 86 mm, spanning an area of 47.3 cm$^2$.

The transducers delivered ultrasound through the parietal and temporal bones of ex-vivo skulls. Specifically for each subject, the transducers were orientated in parallel to the left and right sides of the skull. The transducers were driven by a programmable system (Vantage256, Verasonics).

The second, experimental system was used to test the robustness of RTT with respect to larger array apertures (Suppl. Fig. 3b). In this system, each array had 128 elements (8 x 16 grid) with inter-element spacing of 3 mm on a ellipsoid ($R\_x = 100$ mm, $R\_y = 120$ mm,

$R\_z = 165$ mm). Each array had a height of 74 mm and a width of 151 mm, spanning an area of 105.6 cm$^2$ (Suppl. Fig. 3b).

**Targeting**. Targeting with ultrasound rests on emitting ultrasound from each element such that the wavefronts arrive into the defined target at the same time. These values can be established using 1) the knowledge of the distance from target to the transducer elements 2) measurements using a hydrophone. We used the second approach to measure the arrival time at a specific target in the skulls and the delays that allow the elements to arrive in phase at the same time. The delays are measured by the hydrophone in free-field. With the hydrophone at the target, each element of both transducers is fired individually and its waveform recorded by the hydrophone. The time of flight of each waveform is then measured as the time from when the waveform was emitted by transducer to the time when the waveform arrived at the hydrophone. We then applied delays to the elements that were equal to their time of flight such that they arrive at the target perfectly in phase. In these measurements, each element was driven with 10 cycles of a 650 kHz sine wave with an amplitude of 15 V.

**Correction methods**. This study specifically focuses on the correction of the ultrasound attenuation, which is the key factor in delivering controlled ultrasound intensity into the brain. The goal is to measure the attenuation and compensate for it, ideally as if no skull was present. The method also measures the time of flight and so enables also the correction for the less relevant dephasing of the ultrasound.

The signal emitted from each transducer $i$ on its path to specific brain target of interest is attenuated by acoustic obstacles (skull, hair, coupling) by a factor of $A_i$, and sped up by $\tau_i$. Each factor $A_i$ and $\tau_i$ is specific to the position of the target due to its unique path through the skull. The aim of the below correction methods is to estimate these values and compensate for them. The compensation scales the amplitude of each beam by a factor of $\frac{1}{A_i}$, and delays the emission time by $\tau_i$. Critically, in this method, the measurements obtained through the skull are compared to reference measurements obtained in water.

*No correction*. No adjustments to the emission times and amplitudes were performed following the targeting in water.

*Hydrophone correction*. This correction uses a hydrophone positioned at the target to measure $A_i$ and $\tau_i$ directly. These measurements provide the hypothetical ground truth and serve as a benchmark. The relative attenuation, $A_i$, is measured as the ratio of the peak negative pressure of the two waveforms. The peak negative pressure was computed as the median of the negative cycle peaks over the 10 cycles. The relative speed-up time, $\tau_i$, was obtained as the time that maximizes the cross-correlation between the waveform received through the skull and in free field.

*Relative through-transmit correction*. In this method, the transducers sequentially emitted a 10-cycle, 650 kHz pulse from each individual element while recording responses from all the other, nontransmitting elements (Suppl. Fig. 4). During the through-transmit scans, the peak pressure amplitude of each transducers was 80 kPa. The entire process of this scan takes less than 1 s to complete. The 650 kHz pulse frequency is the same as that used for the neuromodulation. The equivalence of energy kind (acoustic) and frequency (650 kHz) between the through-transmit measurement and the neuromodulatory ultrasound enables a direct measurement of the ultrasound attenuation and

phase shift by the skull and other obstacles in the path, compared to indirect imaging methods such as CT or MRI. This through-transmit measurement is relativistic, performed in comparison to reference measurements that had been taken in water for the same, fixed geometry. The relative differences in the received ultrasound waveforms between the two conditions enable the computation of $\frac{1}{A_i}$ and $\tau_i$ (Suppl. Fig. 2). The $A_i$ and $\tau_i$ values are computed separately.

*Relative through-transmit; correction for attenuation*. The determination of the attenuation of each ultrasonic beam amounts to solving the following system of equations:

$$\ln A_{ij} = \ln A_i + \ln A_j,$$

where $A_{ij}$ is the relative attenuation measured by the through-transmit method for ultrasound propagating from element $j$ to element $i$, through both sides of the skull (Fig. 2). The attenuation values through the two opposite segments of the skull are multiplicative; hence the logarithmic formulation for attenuation. This linear system of equations can be represented in a matrix form as $Mx = b$, where $M$ is a matrix of the unitary value coefficients for transmit-receive pairs, $x$ is a vector of the sought values ($x = [\ln A_1, \ln A_2, \dots, \ln A_{256}]$), and $b$ is a vector of the measured values ($\ln A_{ij}$).

This inverse problem can be solved using a variety of methods, including ordinary least squares, truncated singular-value decomposition, and Tikhonov regularization. We applied SVD decomposition on the M matrix, inverted to solve for $b$, and used Tikhonov regularization to remove high-frequency noise. This method maximized accuracy. The matrix $M$ was conditioned in three ways, each of which improves the correction accuracy. First, we selected through-transmit pairs in a target-dependent manner. Specifically, we selected pairs where the angle between the transmitting transducer and the target and the transmitting and receiving transducers was less than or equal to 10°. This angle was chosen as a compromise between maximizing addressable space while minimizing the incidence angle to the skull and thus undesirable beam aberrations. The method was relatively insensitive to this choice; values between 8 and 15° provided adequate compensation. Formally, this selection can be written as $WMx = Wb$, where $W$ is a diagonal weighting matrix with diagonal values $w_{ij}$ for each pair transmit-receive elements. Second, we only used information of elements that provided detectable and plausible values ($0.01 \leq A_{ij} \leq 0.85$). And third, to ensure invertability, we add to the system of equations $Mx = b$ an additional set of equations that provide initial values for each $x_i$. These equations are $M_{ii}x_i = \ln\sqrt{A_{ij}}$, where $M_{ii} = 1$ and $j$ is the opposing element of $i$ with the smallest angle to the target point. The Tikhonov's regularization parameter, $\lambda$, was set automatically using generalized cross-validation[59]. A standard machine running Matlab provides the solution $x$ in less than a minute.

*Relative through-transmit; correction for dephasing*. The following text describes the algorithm that compensates for the distortion of the ultrasound phase by the skull. The correction for phase is not key for the purpose of this study but provides additional benefit in terms of the delivery of controlled dose (Suppl. Fig. 1). The sought phase delays applied to each element are denoted as $\boldsymbol{\tau} = [\tau_1, \tau_2, \dots, \tau_N]$. Let $s_{ij}(t)$ represent the signal received on a transducer $i$ after a brief, 10-cycle pulse is emitted from a transducer $j$. Let the signals received in free field and through the acoustic barriers (skull, hair, coupling, and other barriers) be denoted as $s_{ij}^F(t)$ and $s_{ij}^B(t)$, respectively.

The procedure reduces the complexity of all possible combinations of transmit-receive pairs to a simpler case in which

a set of transducers focuses onto a single receiver. This way, the speedup experienced by the skull in front of each receiver is a simple linear combination of the through-transmit measurements. Accordingly, the procedure starts by focusing the transmitting elements onto each receiving element in free field. The vector $\boldsymbol{\tau}_i^F = [\tau_{i1}^F, \ldots, \tau_{ij}^F, \ldots, \tau_{iN}^F]$ defines the delays in free-field that focus the transmitting signals, $s_{ij}^F(t), 1 \leq j \leq N$, onto an element $i$ such that the signals arrive perfectly in phase. Additionally, for each element $i$, we select the set of neighboring elements immediately bordering the element, $\mathbb{E}_i$. For all pairs of elements $i$, and $j \in \mathbb{E}$, we select the intersection of receive elements, $\mathbb{R}$, with angle less than 10° between the transmitting element and target, and transmitting element and the receive element. The vector $\boldsymbol{\delta\tau}_{ij}^F = [\tau_{ij1}^F, \ldots, \tau_{ijr}^F, \ldots, \tau_{ijR}^F]$ defines the difference in phase in free field from transmitting elements $i$ and $j$ to each receive element $r$.

Let us define the set of signals received by the element $i$ from all transmitting elements as $\boldsymbol{s}_i = [s_{i1}(t), s_{i2}(t), \ldots, s_{iN}(t)]$. As with the attenuation correction, we control which through-transmit pairs contribute to the equation by including a weighting function $\boldsymbol{w}_i(\theta, p) = [w_{i1}, w_{i2}, \ldots, w_{iN}]$, where $p$ is the target location and $\theta$ is the acceptance angle of the angle between transducer element $i$ and the target and transducer element $i$ and the receive transducer. We set $\theta = 10°$ and let $w_{ij} = 1$ for elements with angle below the acceptance angle and $w_{ij} = 0$ otherwise, with the same argument as for the attenuation. Let the sum of the received signals, for any vector of delays $\boldsymbol{\tau}$, and any set of weights, $\boldsymbol{w}_i$, be denoted as $S(\boldsymbol{s}_i, \boldsymbol{\tau}, \boldsymbol{w}_i, t) = \sum_j^N w_{ij} s_{ij}(t) * \delta(t - \tau_j)$.

The goal is to find the delays $\boldsymbol{\tau}$ that account for the speed-up through the obstacles (skull) relative to free-field (water). The focused signal received by element $i$ in water, $S(\boldsymbol{s}_i^F, \boldsymbol{\tau}_i^F, \boldsymbol{w}_i, t)$, should be delayed by $\tau_i$ compared with the signal received through through the skull after applying delays $\boldsymbol{\tau}$ to all other transmitting elements to compensate for their respective speedup due to the skull, i.e., $S(\boldsymbol{s}_i^B, (\boldsymbol{\tau}_i^F + \boldsymbol{\tau} + \overrightarrow{1}\tau_i), \boldsymbol{w}_i, t)$. The delays, $\boldsymbol{\tau}$, are found by maximizing the coherence between the through-water and through-skull signals. This amounts to maximizing the following criterion:

$$C(\boldsymbol{\tau}) = \sum_{i=1}^N \left( r\left(S\left(\boldsymbol{s}_i^F, \boldsymbol{\tau}_i^F, \boldsymbol{w}_i, t\right), S\left(\boldsymbol{s}_i^B, \left(\boldsymbol{\tau}_i^F + \boldsymbol{\tau} + \overrightarrow{1}\tau_i\right), \boldsymbol{w}_i, t\right)\right) \right.$$
$$\left. + r\left(\sum_j^{\mathbb{E}_i} S\left(\boldsymbol{s}_j^B, \boldsymbol{\delta\tau}_{ij} + \overrightarrow{1}\tau_j, \boldsymbol{w}_i, t\right), S\left(\boldsymbol{s}_i^B, \left(\boldsymbol{\tau}_i^F + \boldsymbol{\tau} + \overrightarrow{1}\tau_i\right), \boldsymbol{w}_i, t\right)\right) \right)$$

where $r$ denotes the Pearson's correlation.

This optimization problem can be effectively solved through an iterative approach. Starting with $\boldsymbol{\tau} = \overrightarrow{0}$ for $n = 0$, for each element $i$, we estimate $\tau_i$ such as to maximize the correlation

$$r(S(\boldsymbol{s}_i^F, \boldsymbol{\tau}_i^F, \boldsymbol{w}_i, t), S(\boldsymbol{s}_i^B, (\boldsymbol{\tau}_i^F + \boldsymbol{\tau} + \overrightarrow{1}\tau_i), \boldsymbol{w}_i, t)) +$$
$$r(\sum_j^{\mathbb{E}_i} S(\boldsymbol{s}_j^B, \boldsymbol{\delta\tau}_{ij} + \overrightarrow{1}\tau_j, \boldsymbol{w}_i, t), S(\boldsymbol{s}_i^B, (\boldsymbol{\tau}_i^F + \boldsymbol{\tau} + \overrightarrow{1}\tau_i), \boldsymbol{w}_i, t))$$

After iterating through each individual element, the delay for each is set to the currently estimated $\tau_i$. The process is then repeated. This iterative procedure terminates when the criterion $|C|$ converges to $|C(\boldsymbol{\tau}^{n+1}) - C(\boldsymbol{\tau}^n)| < 0.1$ or after 10 iterations. This stopping rule provides a favorable trade-off between compensation accuracy and computation time. For example, a standard machine running Matlab provides a solution in approximately 4 minutes for this stopping rule.

Notably, the keeping track of all $s_{ij}^F(t)$ and $s_{ij}^B(t)$ and optimizing the above criterion is necessary to avoid attempts to make exact predictions of phase, which are vulnerable to cycle skipping. In

cycle skipping, signals that are seemingly in phase are in fact off by a multiple of the period. This could occur if the phase shifts between the free-field and through-skull measurements were computed directly. The method instead attempts to predict time offsets such that waves arrive at the target in phase, regardless if they are off by an integer multiple of the period. We also tested how the phase correction performs when the correction is performed without a regard to a specific target. To do so, the weight vector is set to include phase information of all elements. (The amplitude correction in this case still uses a 10 degree acceptance angle to exclude remote elements, but now around the normal vector of each element.) The correction is calculated once and applied to all targets. This test did not have a substantial impact on the results Suppl. Fig. 8.

**Skulls.** Eight ex-vivo human skulls were used in this study. The skulls were obtained from Skulls Unlimited (Oklahoma City, OK). The supplier provides ex-vivo specimens specifically for research under a research agreement. A large opening was made at the bottom of each skull to enable field measurements inside the skull. Each skull was degassed overnight in deionized water[20]. Following the degassing at -25 mmHg, the skull was transferred, within the degassed water, into an experimental tank filled with continuously degassed water (AIMS III system with AQUAS-10 Water Conditioner, Onda).

**Hydrophone field scans.** A capsule hydrophone (HGL-0200, Onda) secured to 3-degree-of-freedom programmable translation system (Aims III, Onda) was used to record the ultrasound field emitted from each element. The hydrophone has a sensitivity of -266 dB relative to 1 V µPa$^{-1}$ and aperture size of 200 µm. This aperture size is well within the ultrasound wavelength (2.3 mm). The 3D field measurements use a step size of 0.2 mm to provide high spatial resolution of each element's contribution to the total field. The hydrophone scans traversed a volume of 5 x 5 x 5 mm for each target. We also performed wider, 10 x 10 mm planar (XY and YZ) scans for each target. The fields for all skulls and dimensions are detailed in Suppl. Fig. 7.

The scans were performed in free-field (reference scan) and through each ex-vivo skull. At each location in the scans, elements were fired individually, and the received signals recorded. Since ultrasound pressure is additive, the total pressure was computed as the sum of the individual constituents. We measured the spatial peak intensity of the entire field at each target by taking the maximum intensity value in the measured volume. Position error of this peak and focal volume are quantified in Suppl. Fig. 9.

**RTT in human subjects.** The hardware and approach described in this article was considered nonsignificant risk by the Institutional Review Board of University of Utah and approved to be applied in healthy individuals (Protocol #00127033; no ultrasonic stimulation) and patients with major depression in conjunction with ultrasonic stimulation (Protocol #00148802; preregistered NCT05301036). All subjects provided informed consent. Participants were 4 healthy subjects (4 males, aged between 25 and 40 years; data points 1-4 in Fig. 5) and 2 subjects with major depression (female, 35 years; female, 32 years). No hair shaving was necessary as RTT takes hair and other obstacles within the ultrasound path into account. No subject was excluded.

Each subject had the two-phased array transducers placed parallel over the left and right sides of their head. Coupling was mediated using a hydrogel[60]. Standard ultrasound coupling gel was applied to the interfaces between the transducer and the hydrogel, and the hydrogel and the head. The RTT scan was performed

through the entire system of head and coupling. No subject reported detrimental effects during or following the procedure.

## Deep brain stimulation

*Stimulation parameters.* We validated RTT performance by delivering neuromodulatory ultrasound into a deep brain target, the subgenual cingulate cortex (Fig. 4). During the stimulation and imaging, the patient's head was immobilized with a standard radiological thermoplastic mask. The transducer arrays were registered to the subject inside the MRI using fiducial markers. RTT was performed in the same manner as in the healthy individuals (see above). The ultrasound was delivered into the target (subgenual cingulate cortex) in 30 ms pulses (650 kHz, 1.0 MPa peak pressure) every 4 s. The stimulation was administered in 1-minute ON blocks, followed by 1-minute OFF blocks of no ultrasound, for a total of up to 10 min. The frequency of 650 kHz[18] was chosen as a compromise between higher frequencies that provide sharper focus and lower frequencies that are less attenuated by the skull[18]. The pressure and pulsing parameters were chosen to be similar to previous ultrasound neuromodulation studies[4,5,9] while staying within the FDA 510(k) Track 3 guidelines[31]: $I_{SPTA} = 0.23$ W cm$^{-2}$ and $I_{SPPA} = 31.0$W cm$^{-2}$.

*Imaging acquisition and analysis.* The fMRI BOLD scan (Fig. 4) used standard T2$\star$-weighted sequence: interleaved series, P-A phase encoding, TR 2.0 s, TE 33 ms, FA 80 degrees, FOV 207 mm, 52 slices, slice thickness 2.4 mm, bandwidth 2004 Hz/pixel, echo spacing 0.62 ms, 300 volumes per 10 minutes. The acquired fMRI data were processed using SPM12 (RRID:SCR_007037) and ANIMA (RRID:SCR_017017) packages. The processing consisted of four standard steps: i) co-registration of anterior to posterior and posterior to anterior field map to time-series (ANIMA) ii) echo-planar imaging (EPI) distortion correction (ANIMA) iii) realignment of time-series data (SPM12), and iv) application of Gaussian smoothing using a 8 mm kernel (SPM12). Significance was determined using a false discovery rate correction with a p-value of < 0.001. Minimum cluster size was set at 30 voxels. Standard general linear model regressed the stimulation factor (i.e., the blocks of 1-minute ON and 1-minute OFF stimulation) on the BOLD activity. The statistical difference between the ON and OFF outputs was assessed using a *t*-test.

## Assessments of skull temperature

*Simulations.* The pressure field of the arrays was simulated using Field II[61]. We applied the RTT correction values used in human subjects (Fig. 4). To measure the peak temperature rise in the skull, we calculated skull heating using the maximal pressure in the range of a subject's skull, 0-50 mm from the transducer face. We computed the temperature increase using the bioheat equation, $\Delta T = \frac{2\alpha I \Delta t}{\rho C}$, where $\alpha$ is the absorption of the skull bone $(1.5f^{1.7} = 1.5(0.65)^{1.7} = 0.63$ m$^{-1})$, $C$ is the specific heat of the skull (1300 J kg$^{-1}$ K$^{-1}$), $\rho$ is the density of skull bone (1700 kg m$^{-3}$), $I$ is the spatial peak pulse average intensity in skull bone (2.2 W cm$^{-2}$), and $\Delta t$ is the pulse duration. The equation does not include heat conduction and convection and therefore provides an upper bound estimate on the temperature rise.

*Measurements.* We measured temperature inside three ex-vivo skulls using a fiber optic hydrophone (FOHS, Precision Acoustics). For each skull, the fiber was embedded 1-2 mm into the depth of each skull. The arrays were then mounted over the left and right sides of the skull and the setup submerged in a water tank. The ultrasound transducer array was scanned in a 10 x 10 x 10 mm grid, which provided the location of the maximum pressure and thus the worse case scenario. We then applied the RTT correction that was used in the most attenuating human subject (Fig. 4) and sonicated the same central target inside the skull with the according pressure used in the subject, i.e., 1 MPa. The ultrasound was delivered with the same inter-stimulus interval as in the subject, i.e., every 4 s. The total duration of the insonation was 1 minute. We varied pulse duration from 10 to 300 ms and measured the immediate change in temperature for each pulse duration.

**Reporting summary.** Further information on research design is available in the Nature Portfolio Reporting Summary linked to this article.

## Data availability

The data associated with the measurements are provided in the article. For raw data, contact the corresponding author.

## Code availability

The code associated with the article is available upon request.

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

## Acknowledgements

This work was supported by the NIH grants R00NS100986, RF1NS128569, S10OD026788, the Ascender grant (University of Utah Technology Transfer Office), the Mildred Foundation, and the University of Utah College of Engineering seed grant. We thank Dr. Richard Rabbitt, Dr. Douglas Christensen and Dr. Dennis Parker for helpful comments. We thank John Rolston for the assessment of a skull with a potential hyperostosis. We thank Lily Vonesh for scheduling the subjects.

## Author contributions

J.K. developed the original concept. T.R. performed the research. D.F. processed the MRI data. B.M. and T.R. performed the stimulation in patients.

## Competing interests

The approaches and hardware described herein are subject to a pending patent of which J.K. holds a competing interest.

## Ethical Statement

This research was conducted in accordance with the principles embodied in the Declaration of Helsinki and in accordance with local statutory requirements. The hardware and stimulation described in this article was considered non-significant risk by the Institutional Review Board of University of Utah and approved to be applied in patients with major depression (Protocol #00148802) and in healthy individuals (Protocol #00127033). All subjects provided informed consent.
