## [Peer Review File · Communications Engineering]

Reviewers' comments:

Reviewer #2 (Remarks to the Author):

This manuscript proposed the relative through-transmit method, which enables direct measurements of the attenuation of the skull, the hair, and the coupling between the scalp and the transducers. The authors identified the method in ex-vivo and in vivo experiments. No CT or MRI images of the head are required, which has significant application value. Some issues need to be further addressed before the manuscript is accepted.

1. RTT compensates for the measured aberrations and delivers into a transcranial target a defined level of ultrasound intensity. This means an increase in the energy emitted by ultrasound, which may lead to a significant increment of cranial heat due to the absorption of skull. This should be discussed in discussion.
2. This manuscript seems to be solely focused on RTT and phase correction. However, there is a lack of comparison between the results of this work and what has been achieved with other method, such as time reversal. Making some compared to the time reversal would be helpful.
3. In human subjects, the ultrasound stimulation parameters (i.e. frequency, duty cycle, pulse repetition frequency, acoustic pressure) have not been justified. Being a manuscript on demonstrating the use of ultrasound to achieve neuromodulation, the choice of ultrasound parameters needs to be scientifically explained.
4. The authors mentioned that ultrasound stimulation is safe, but do not provide evidence. It would be a good idea to perform MR imaging to demonstrate the safety of ultrasound stimulation.
5. Conclusion: This single-sentence section is poorly developed and needs to be extended. The authors can add some foresight about this technology.

Reviewer #3 (Remarks to the Author):

The subject of this manuscript is very meaningful, and the proposed method seems interesting for the researchers who conduct investigations of ultrasound neuromodulation. To help the authors enhance their manuscript, I have the following technical comments for them:

1. How did you ensure the two transducer arrays on the opposite side of the skull target to the same point in the skull?
2. The cross-section of human skull supposes to be nonsymmetric, how did you ensure the ultrasonic beam was not distorted?
3. Did you measure the safety level of the proposed method, such as temperature rise inside the tissue of skull?
4. Why was the transducer frequency of 650 kHz selected? Did the authors consider higher frequencies, such as 1.5 MHz and 2 MHz?
5. How did the authors suggest to improve this method for clinical applications?

We thank the Reviewers for their time and attention in reviewing this manuscript. We have addressed the constructive criticisms in detail below.

Reviewer #2 (Remarks to the Author):

This manuscript proposed the relative through-transmit method, which enables direct measurements of the attenuation of the skull, the hair, and the coupling between the scalp and the transducers. The authors identified the method in *ex-vivo* and *in vivo* experiments. No CT or MRI images of the head are required, which has significant application value. Some issues need to be further addressed before the manuscript is accepted.

1. RTT compensates for the measured aberrations and delivers into a transcranial target a defined level of ultrasound intensity. This means an increase in the energy emitted by ultrasound, which may lead to a significant increment of cranial heat due to the absorption of skull. This should be discussed in discussion.

#Authors' Response:

The reviewer raises an excellent point about the safety concerns of skull heating. This issue was addressed in the manuscript previously. Specifically, we both simulated and measured the skull heating that results from the stimulation following the RTT correction.

To make these findings more explicit, the previous Supplementary Figure 13 is now Figure 5 in the main text.

The Results now include the Fig. 5 (below)

as well as the accompanying text:

“We also confirmed that the method did not lead to appreciable heating of the skull. Specifically, we performed simulations and measurements of the temperature rise inside 3 *ex-vivo* skulls (Methods), following the application of RTT. In both the simulation and the measurements, the ultrasonic stimuli matched the 1 MPa pressure amplitude used in the SGC stimulation (Fig. 4). Fig. 5 shows the simulated and measured peak temperature rise due to

ultrasonic stimulation as a function of pulse duration. The pulse duration used in the stimulation of the subjects (30 ms; Fig. 4) led to a maximum of 0.047°C temperature increase across all three skulls tested. The simulations and the measurements did not incorporate heat convection by blood vessels, and therefore likely represent upper-bound estimates of the temperature increase.”

Additionally, we expanded the respective Discussion paragraph to:

“The compensation for attenuation could conceivably heat the skull during the ensuing low-intensity application. To address this issue, we calculated and measured the peak skull heating inside ex-vivo human skulls for the respective neuromodulation parameters and found no concerning levels of temperature rise (Fig. 5). This is expected as the overall energy deposited into the skull for low intensity therapies, after RTT correction, is orders of magnitude lower than the intensities that can produce harmful skull heating⁵⁶. Nonetheless, in every case, including after the application of RTT, it is recommended to pulse the ultrasound such as to follow the I_{SPTA} limit of the FDA 510(k) Track 3 guidelines³¹.”

2. This manuscript seems to be solely focused on RTT and phase correction. However, there is a lack of comparison between the results of this work and what has been achieved with other method, such as time reversal. Making some compared to the time reversal would be helpful.

#Authors’ Response:

We addressed this comment by expanding the Discussion by five studies that used time reversal:

“Critically, in RTT, the acoustic path is measured entirely noninvasively. In comparison, time-reversal methods use implanted receivers^{32, 33} or injected microbubbles^{34–36}. Although these methods can provide accurate corrections, their invasiveness has limited their deployment in humans.”

3. In human subjects, the ultrasound stimulation parameters (i.e. frequency, duty cycle, pulse repetition frequency, acoustic pressure) have not been justified. Being a manuscript on demonstrating the use of ultrasound to achieve neuromodulation, the choice of ultrasound parameters needs to be scientifically explained.

#Authors’ Response:

In response to this comment, we now add justification for our ultrasound parameter choice for neuromodulation in humans.

In Methods:

“The ultrasound was delivered into the target (subgenual cingulate cortex) in 30 ms pulses (650 kHz, 1.0 MPa peak pressure) every 4 s. The stimulation was administered in 1-minute ON blocks, followed by 1-minute OFF blocks of no ultrasound, for a total of up to 10 min. The frequency of 650 kHz was chosen as a compromise between higher frequencies that provide sharper focus and lower frequencies that are less attenuated by the skull¹⁸. The pressure and

pulsing parameters were chosen to be similar to previous ultrasound neuromodulation studies^{4,5,9} while staying within the FDA 510(k) Track 3 guidelines³¹: $I_{SPTA} = 0.23 \text{ W/cm}^2$ and $I_{SPPA} = 31.0 \text{ W/cm}^2$.

4. The authors mentioned that ultrasound stimulation is safe, but do not provide evidence. It would be a good idea to perform MR imaging to demonstrate the safety of ultrasound stimulation.

#Authors' Response:

We have addressed this comment in the following ways:

- 1) We highlighted the measurements of skull heating, which show a negligible temperature rise, by presenting this figure (Fig. 5) in the main text, while accordingly expanding the discussion
- 2) In Table S1, we provide the subjects' responses to surveys of 34 potential side effects, which noted no adverse events related to treatment.
- 3) We now include a new Supplementary Figure 13, attached also below, in the manuscript. The figure shows MRI images of the SGC before and after the stimulation. There are no detectable signs of damage.

4)

Our previous study⁵⁷ performed histology in non-human primates and sheep at exposures levels higher than those used in this study. No histological findings were detected⁵⁷. The Discussion now includes the following text:

“ RTT was applied to the human head safely (Suppl. Fig. 5, Fig. 4, Suppl. Table S1, Suppl. Fig. 13). RTT uses diagnostic- imaging-like, $< 100 \mu\text{s}$ pulses that were safely within the FDA 510(k) Track 3 guidelines³¹: $I_{SPTA} = 5.4 \text{ mW/cm}^2$ and $I_{SPPA} = 1.3 \text{ W/cm}^2$. Following the correction, there were no reported side effects in either subject in response to stimulation of the subgenual cingulate cortex (Suppl. Table S1). The compensation for attenuation could conceivably heat the skull during the ensuing low-intensity application. To address this issue, we calculated and measured the peak skull heating inside ex-vivo human skulls for the respective neuromodulation parameters and found no concerning levels of temperature rise (Fig. 5). This is expected as the overall energy deposited into the skull for low intensity therapies, after RTT correction, is orders of magnitude lower than the intensities that can produce harmful skull heating⁵⁶. Nonetheless, in every case, including after the application of RTT, it is recommended to pulse the ultrasound such as to follow the I_{SPTA} limit of the FDA 510(k) Track 3 guidelines³¹. Furthermore, a previous study⁵⁷ performed histology in non-human primates and sheep at exposure levels higher than those used in this study. No histological findings were detected⁵⁷. Finally, no tissue damage was detected in T1-weighted MRI images of either subject following the stimulation (Suppl. Fig. 13).”

5. Conclusion: This single-sentence section is poorly developed and needs to be extended. The authors can add some foresight about this technology.

#Authors' Response:

We addressed this comment in two ways:

- 1) We expanded the concluding paragraph to better summarize on the strengths of the technology: “In summary, we developed and deployed in humans a noninvasive approach that accurately and safely compensates for the severe and unpredictable attenuation of ultrasound by the head. The approach delivers controlled ultrasound intensity through the human skull and enabled targeted modulation of deep brain circuits in humans. This practical correction method is deployed in subjects in real time, does not require CT or MRI scans of the head, and accounts for all obstacles in the ultrasound path, thus circumventing the need for hair shaving. This way, the method is expected to be applicable to a broad spectrum of subjects and patients, enabling precise and personalized diagnoses and treatments of the brain.”

- 2) We expanded the discussion by future directions for the technology in clinical settings: “Clinical applications will benefit from the method's rapid deployment, which enables to check coupling quality before and many times during an ultrasound delivery session, thus accounting for subjects' possible movements introduction of air gaps or bubbles along the beam path To maximize the effectiveness and safety of clinical applications, additional work on the compensation could further tighten the confidence on the delivered intensity. This would allow operators to deliver higher

intensities without the risk of exceeding safety limits. Clinical applications would also benefit from MRI-based measurements of the in-situ intensities using acoustic radiation force imaging³⁸.”

Reviewer #3 (Remarks to the Author):

The subject of this manuscript is very meaningful, and the proposed method seems interesting for the researchers who conduct investigations of ultrasound neuromodulation. To help the authors enhance their manuscript, I have the following technical comments for them:

1. How did you ensure the two transducer arrays on the opposite side of the skull target to the same point in the skull?

#Authors' Response:

In response to this comment, we have expanded the methods to clarify this important point.

In Methods:

“Targeting with ultrasound rests on emitting ultrasound from each element such that the wavefronts arrive into the defined target at the same time. These values can be established using 1) the knowledge of the distance from target to the transducer elements 2) measurements using a hydrophone. We used the second approach to measure arrival time at a specific target in the skulls and the delays that allow the elements to arrive in phase at the same time. The delays are measured by the hydrophone in free-field. With the hydrophone at the target, each element of both transducers is fired individually and its waveform recorded by the hydrophone. The time of flight of each waveform is then measured as the time from when the waveform was emitted by transducer to the time when the waveform arrived at the hydrophone. We then applied delays to the elements that were equal to their time of flight such that they arrive at the target perfectly in phase. In these measurements, each element was driven with 10 cycles of a 650 kHz sine wave with an amplitude of 15 V.”

2. The cross-section of human skull supposes to be nonsymmetric, how did you ensure the ultrasonic beam was not distorted?

#Authors' Response:

Indeed, the ultrasound beam is distorted by the skull. We have addressed this issue in the following ways:

- 1) The developed hardware delivers the ultrasound through the parietal and temporal bones. Previous studies (Lindsey et al. 2013, Ammi et al. 2008, Riis et al. 2021) found**

that delivery of the ultrasound through the temporal and parietal bone minimizes the severity of the aberrations.

- 2) These areas of the human skull have relatively low curvature. We designed the geometry of our arrays to minimize the incidence angle with the skull over these areas of the head.
- 3) Supplementary Fig. 7 shows that the distortion following RTT is relatively small with respect to the beam width.
- 4) The RTT algorithm makes no assumption about the symmetry of the skull and calculates the distortion in front of each element individually.
- 5) RTT operates on relative measurements, i.e., measurements performed through the skull versus free field. The goal is to adjust the through transmit measurements through the skull to match the through transmit measurements through free field with amplitude and phase shift corrections, regardless of the source of the distortion (shifting beam, attenuation, speedup through the skull, reflection, presence of hair or air bubbles, etc.).

These points are now incorporated in the Discussion:

“We chose to place transducers parallel over the left and right sides of the skull as i) there is a through-transmit path with minimal incidence angle to the skull ii) the parietal bone has favorable acoustic properties¹⁹, and iii) this geometry provides convenient coupling.”

“No such assumption can be made for RTT, which uses large, therapeutic arrays and solves for the distortions in front of each individual element in a system of equations containing all transducer elements.”

“RTT implements a virtual line path between each transmit-receive pair of elements, and thus measures all forms of attenuation (reflection, absorption, and scattering) of all obstacles along this defined path (Suppl. Fig. 2). Conceptually, RTT could be considered a kind of ultrasound computed tomography⁵¹⁻⁵⁵. Unlike tomography, however, RTT compares signals with the head present to reference signals acquired in water (free field), thus specifically determining the relative attenuation of each ultrasonic beam. The method does not aim to provide absolute attenuation values for each voxel of the tissue, which would be a much more difficult problem.”

3. Did you measure the safety level of the proposed method, such as temperature rise inside the tissue of skull?

Yes, this important issue was addressed in the manuscript previously. Specifically, we both simulated and measured the skull heating that results from the stimulation following the RTT correction.

To make these findings more explicit, the previous Supplementary Figure 13 is now Figure 5 in the main text.

The Results now include the Fig. 5 (below)

as well as the accompanying text:

“We also confirmed that the method did not lead to appreciable heating of the skull. Specifically, we performed simulations and measurements of the temperature rise inside 3 *ex-vivo* skulls (Methods), following the application of RTT. In both the simulation and the measurements, the ultrasonic stimuli matched the 1 MPa pressure amplitude used in the SGC stimulation (Fig. 4). Fig. 5 shows the simulated and measured peak temperature rise due to ultrasonic stimulation as a function of pulse duration. The pulse duration used in the stimulation of the subjects (30 ms; Fig. 4) led to a maximum of 0.047°C temperature increase across all three skulls tested. The simulations and the measurements did not incorporate heat convection by blood vessels, and therefore likely represent upper-bound estimates of the temperature increase.”

Additionally, we expanded the respective Discussion paragraph to:

“The compensation for attenuation could conceivably heat the skull during the ensuing low-intensity application. To address this issue, we calculated and measured the peak skull heating inside *ex-vivo* human skulls for the respective neuromodulation parameters and found no concerning levels of temperature rise (Fig. 5). This is expected as the overall energy deposited into the skull for low intensity therapies, after RTT correction, is orders of magnitude lower than the intensities that can produce harmful skull heating⁵⁶. Nonetheless, in every case, including after the application of RTT, it is recommended to pulse the ultrasound such as to follow the I_{SPTA} limit of the FDA 510(k) Track 3 guidelines³¹.”

4. Why was the transducer frequency of 650 kHz selected? Did the authors consider higher frequencies, such as 1.5 MHz and 2 MHz?

#Author’s Response:

In response to this comment, we now add justification for our transducer center frequency choice in methods. We also include discussion of the frequency choice as a limitation of the

method, namely that higher frequencies cannot pass through both sides of the skull.

In Methods:

“The ultrasound was delivered into the target (subgenual cingulate cortex) in 30 ms pulses (650 kHz, 1.0 MPa peak pressure) every 4 s. The stimulation was administered in 1-minute ON blocks, followed by 1-minute OFF blocks of no ultrasound, for a total of up to 10 min. The frequency of 650 KHz was chosen as a tradeoff between higher frequencies that provide sharper focus and lower frequencies that are less attenuated by the skull¹⁸. The pressure and pulsing parameters were chosen to be similar to previous ultrasound neuromodulation studies^{4,5,9} while staying within the FDA 510(k) Track 3 guidelines³¹: ISPPA = 31 W/cm² and ISPTA = 0.233 W/cm².”

In Discussion:

“Second, the method may be limited to frequencies below 2 MHz, which can penetrate both sides of the skull⁴⁵. This is generally not an issue for transcranial therapeutic ultrasound applications, which typically use frequencies below 1 MHz to provide adequate penetration of the skull¹⁸.”

5. How did the authors suggest to improve this method for clinical applications?

#Authors' Response:

In response to this comment, we have added a paragraph to the Discussion suggesting improvements to the method for clinical applications.

In Discussion:

“Clinical applications will benefit from the method's rapid deployment, which enables to check coupling quality before and many times during an ultrasound delivery session, thus accounting for subjects' possible movements introduction of air gaps or bubbles along the beam path. To maximize the effectiveness and safety of clinical applications, additional work on the compensation could further tighten the confidence on the delivered intensity. This would allow operators to deliver higher intensities without the risk of exceeding safety limits. Clinical applications would also benefit from MRI-based measurements of the in-situ intensities using acoustic radiation force imaging³⁸.”